# Effect of Air Classification and Enzymatic and Microbial Bioprocessing on Defatted Durum Wheat Germ: Characterization and Use as Bread Ingredient

**DOI:** 10.3390/foods13121953

**Published:** 2024-06-20

**Authors:** Angela Longo, Gianfranco Amendolagine, Marcello Greco Miani, Carlo Giuseppe Rizzello, Michela Verni

**Affiliations:** 1Department of Environmental Biology, “Sapienza” University of Rome, Piazzale Aldo Moro 5, 00185 Rome, Italy; angela.longo@uniroma1.it (A.L.); carlogiuseppe.rizzello@uniroma1.it (C.G.R.); 2Casillo Next Gen Food s.r.l, Via Sant’Elia, SNC, 70033 Corato, BRI, Italy; gianfranco.amendolagine@casillogroup.it (G.A.); marcello.miani@casillogroup.it (M.G.M.)

**Keywords:** defatted durum wheat germ, lactic acid bacteria, xylanase, bread volume, glycemic index

## Abstract

Its high dietary fiber and protein contents and nutritional quality make defatted wheat germ (DWG) a valuable cereal by-product, yet its negative impact on food structure limits its use as a food ingredient. In this research, DWG underwent air classification, which identified two fractions with high fiber (HF) and low fiber/high protein (LF) contents, and a bioprocessing protocol, involving treatment with xylanase and fermentation with selected lactic acid bacterial strains. The degree of proteolysis was evaluated through electrophoretic and chromatographic techniques, revealing differences among fractions and bioprocessing options. Fermentation led to a significant increase in free amino acids (up to 6 g/kg), further enhanced by the combination with xylanase. When HF was used as an ingredient in bread making, the fiber content of the resulting bread exceeded 3.6 g/100 g, thus reaching the threshold required to make a “source of fiber” claim according to Regulation EC No.1924/2006. Meanwhile, all breads could be labeled a “source of protein” since up to 13% of the energy was provided by proteins. Overall, bioprocessed ingredients lowered the glycemic index (84 vs. 89) and increased protein digestibility (80 vs. 63%) compared to control breads. Technological and sensory analysis showed that the enzymatic treatment combined with fermentation also conferred a darker and more pleasant color to the bread crust, as well as better crumb porosity and elasticity.

## 1. Introduction

Annually, roughly 40 million tons of durum wheat are produced worldwide [1], of which 10% is cultivated in Italy alone, with an associated land usage of approximately 1.24 million hectares [2]. Up to 3% of the kernel comprises wheat germ, the second largest by-product of the milling industry after bran [3]. The removal of wheat germ is necessary to extend the shelf-life of the flour by limiting lipid oxidation in the wheat germ, which rapidly leads to rancidity. The separated wheat germ requires stabilization to prevent oxidation due to lipase activity, which is aimed at prolonging its shelf-life and making it useful as an ingredient. Stabilization can be achieved with enzyme denaturation, acidification, and oil removal through organic solvents or supercritical CO_2_ extraction [3], each one with its pros and cons. For instance, while limiting lipid oxidation is a critical step to control in all extraction methods, supercritical CO_2_ extraction simplifies the oil refining process and eliminates the solvent distillation stage. Furthermore, several applications in the food, pharmaceutical, and cosmetic industries have been proposed for wheat germ oil [3], which involve the generation of a side stream, defatted durum wheat germ cake (DWG), whereas biological acidification and enzyme denaturation allow for the reutilization of DWG, without any further side stream generation.

DWG contains a significant amount of dietary fibers (more than 30% of its mass) and it is made up of approximately 10–30% proteins, which exhibit comparable quality to the reference defined by the FAO/WHO as well as to egg and milk proteins [4]. These features make DWG a potentially valuable food ingredient, which is subject to high functional and nutritional interest. However, despite the numerous attempts to incorporate DWG in food formulations, its negative impact on the food structure is the major factor limiting its large-scale use. For example, supplementations of 1–3% in bread formulations caused higher crumb firmness and chewiness, while higher percentages of addition resulted in smaller specific volumes and lower elasticity compared to control breads, due to the decrease in gas holding capacity [5]. As a result, most of the research on DWG inclusion in food products has focused on cookies or beverages, where the lack of a gluten network or the high fiber content is less detrimental than in leavened products [4].

When the generation of DWG is unavoidable, its reutilization within the food chain, in staple foods like bread, can be considered a promising strategy to valorize its potential while promoting healthy diets, and yet technological issues limit its use as an ingredient. Thus, the development of novel, efficient, and sustainable technologies within a circular economy framework is imperative to offset the shortcomings of DWG and to promote its utilization as an ingredient in staple foods like bread. Recently, a few authors have proposed the incorporation of improvers such as gluten and ascorbic acid, to enhance the rheological properties of bread [6]. However, this approach does not align with the growing demand for clean-label products [7] since ascorbic acid is an additive classified as E300 by EU Reg. 1333/2008 and gluten is another over-processed ingredient to those already used in bread. Indeed, store-bought breads contain dozens of ingredients, which include several types of flour, water, leavening agents, sweeteners, shorteners, improvers, preservatives, vitamins, and so on [8]. Hence, any addition that does not constitute a core ingredient can jeopardize the clean-label status of a product [7]. Fermentation of DWG with selected lactic acid bacteria (LAB), already applied to other cereal by-products thanks to their broad variety of nutritional, functional, and technological advantages, was suggested by Perri et al. [9] as a promising strategy to reduce DWG’s negative impact on the structure and sensory properties of leavened baked goods.

Moreover, although conflicting results have been achieved, it has been suggested that durum wheat milling by-product fractions, as determined by air classification, could improve the nutritional value of bread made thereof without reducing its quality [10].

To the best of our knowledge, none of the previous studies evaluated the effect of air classification on DWG, nor the possibility of combining the use of fiber-degrading enzymes and fermentation to improve the techno-functional potential of DWG fractions. Therefore, based on the above considerations, in this study, micronized DWG flour underwent air classification and a bioprocessing protocol, involving enzymatic treatment with food-grade xylanase and fermentation with selected LAB strains. An integrated approach, including the microbiological and biochemical characterization of DWG fractions and their protein profile, was adopted. Then, raw and bioprocessed fractions were used as ingredients in bread making, and the breads were characterized in terms of their main nutritional and technological features.

## 2. Materials and Methods

### 2.1. Materials and Air Classification

Defatted durum wheat germ (DWG), resulting from wheat germ oil extraction using n-hexane (BP. 68 °C), was kindly provided by Molino Casillo, Corato (BA), Italy. The content of n-hexane in DWG flour, as required by Directive 2009/32/EC, was below 5 ppm.

After oil extraction and solvent removal, DWG was micronized and underwent air classification. More specifically, a micronized flour was obtained from DWG using an impact mill (UPZ 100, Hosokawa-Alpine, Augusta, Germany). The mill speed was set at 15,000 rpm and the feed rate at 3 kg h^−1^. Then, an air classifier (ATP 50, Hosokawa-Alpine, Augusta, Germany) was used to fractionate the micronized DWG flour into coarse and fine fractions (HF and LF, respectively). The speed of the grader wheel was set at 3000 rpm and the air flow rate at 50 m^3^ h^−1^. The volume-based particle size distribution of the raw materials was analyzed by a Mastersizer 3000 (Malvern Panalytical, Malvern, UK) using the dry module.

### 2.2. DWG Characterization

The proteins (total nitrogen × 5.7), lipids, moisture, total dietary fiber, and ash of DWG fractions were determined according to the Approved Methods of the American Association of Cereal Chemists 46–11.02, 30–10.01, 44–01.01, 32–05.01, and 08–01.01 [11]. Available carbohydrates were calculated as follows: [100 − (proteins + lipids + ash + total dietary fiber)].

Mycotoxins levels (aflatoxins, zearalenone, deoxynivalenol, and ochratoxin A) in DWG fractions were analyzed by an HPLC-MS/MS SCIEX model 5500 + (AB Sciex LLC, Framingham, MA, USA) using an isotopically labeled internal standard, according to the L-MI067 rev.0 2020 method, validated by the Italian certification organization Accredia.

### 2.3. Enzymatic and Microbial Bioprocessing

DWG fractions (HF and LF) were submitted to two bioprocessing protocols, that is, fermentation with LAB strains, alone or combined with an enzymatic treatment. *Lactiplantibacillus plantarum* T6B10 and *Fructilactobacillus sanfranciscensis* A2S5, previously selected for their pro-technological performances during DWG fermentation [9], were used to ferment HF and LF. LAB strains were routinely propagated on De Man, Rogosa, and Sharpe (MRS) (Oxoid, Basingstoke, Hampshire, UK) at 30 °C. Before inoculation, they were cultivated until the late exponential phase of growth was reached (ca. 10 h), harvested by centrifugation at 9000× *g* for 10 min at 4 °C, washed twice in 50 mM sterile phosphate buffer (4 °C, pH 7.0), resuspended in tap water, and used to inoculate DWG fractions.

Bioprocessed DWG fractions were prepared by mixing HF and LF flours and tap water with a dough yield (DY) of 350, corresponding to a ratio of DWG flour:water of 29:71, and inoculated with *L. plantarum* T6B10 and *F. sanfranciscensis* A2S5 (final cell density 7 log cfu/g). Fermentation was carried out at 30 °C for 24 h, obtaining f-HF and f-LF.

A second set of samples was obtained by adding (300 mg/100 g of flour) a commercial hydrolytic enzyme preparation with xylanase activity (150,000 XU/g, Bio-Cat Inc., Troy, VA, USA) to *L. plantarum* T6B10 and *F. sanfranciscensis* A2S5. Bioprocessing was carried out at 30 °C for 24 h, obtaining xf-HF and xf-LF.

### 2.4. Characterization of Bioprocessed DWG Fractions

#### 2.4.1. Microbiological Analysis

For the microbiological analysis, 10 g of sample (before and after bioprocessing) was suspended in 90 mL of sterile sodium chloride (0.9%, *w*/*v*) solution and homogenized with a stomacher BagMixer 400 P (Interscience). The LAB cell density was determined on De Man Rogosa and Sharpe (MRS) agar (Oxoid Ltd., Basingstoke, Hampshire, UK), supplemented with 0.01% cycloheximide (Sigma Chemical Co., St. Louis, MO, USA), at 30 °C for 48 h, under anaerobiosis. Yeasts were cultivated on Sabouraud dextrose agar (SDA, Oxoid) supplemented with 0.01% chloramphenicol (Sigma) at 25 °C for 48 h. Total mesophilic aerobic bacteria were enumerated on plate count agar (PCA, Oxoid) supplemented with 0.01% cycloheximide under aerobic conditions at 30 °C for 48 h, and *Enterobacteriaceae* were cultivated on violet red bile glucose agar (VRBGA, Oxoid) at 37 °C for 48 h.

#### 2.4.2. Acidification

Acidification was monitored to determine the pH, total titratable acidity (TTA), and organic acid production. The pH was measured, every 2 h, by a FiveEasy Plus pH meter (Mettler-Toledo, Columbus, OH, USA) with a food penetration probe. TTA, determined before and after DWG bioprocessing, was defined as the amount of 0.1 M NaOH required to adjust the pH of 10 g dough in sterile water to 8.3.

Kinetics of acidification were modeled according to the Gompertz equation as modified by Zwietering et al. [12]—y = k + A exp {−exp[(Vmaxe/A)(λ − t) + 1]}—where y is the acidification extent expressed as ΔpH at the time t; k is the initial level of the dependent variable to be modeled; A is the difference in pH between inoculation and the stationary phase; Vmax is the maximum acidification rate; and λ is the length of the latency phase expressed in hours.

Water/salt-soluble extracts (WSEs) of the doughs were prepared [13] and used to quantify lactic and acetic acids, with K-DLATE and K-ACET kits (Megazyme International Ireland Limited, Bray, Ireland), respectively. The fermentation quotient (FQ), that is, the molar ratio between lactic and acetic acids, was also determined. Megazyme kit K-PHYT 05/07 (Megazyme) was employed, according to the manufacturer’s instructions, for the analysis of phytic acid.

#### 2.4.3. Proteolysis

To assess the effect of enzymatic and microbial bioprocessing on DWG, WSEs, prepared as described above, were also used to evaluate the degree of proteolysis through electrophoretic and chromatographic techniques. The protein patterns were characterized by sodium dodecyl sulfate–polyacrylamide gel electrophoresis (SDS-PAGE) using Bio-Rad 12% Mini-PROTEAN TGX Stain-Free precast gels (Bio-Rad Laboratories GmbH, Feldkirchen, Germany). The Precision Plus Protein™ All Blue and Unstained Protein Standard (Bio-Rad Laboratories) were used as molecular weight markers (10–250 kDa). Gels were run at room temperature for 35 min at 200 V (60 mA, 100 W) in a vertical electrophoresis cell (Bio-Rad Laboratories). Protein bands were visualized using a Gel Doc™ EZ Imager system (Bio-Rad Laboratories) and analyzed using Image Lab software (Bio-Rad Laboratories).

Peptides profiles were characterized by RP-FPLC, using a Resource RPC column and ÄKTA FPLC equipment, with a 20 µL loop and the UV detector operating at 214 nm (GE Healthcare Bio-Sciences AB, Uppsala, Sweden). An elution gradient was performed at a flow rate of 1 mL/min using a mobile phase composed of water (Eluent A) and acetonitrile (CH_3_CN, Eluent B), containing 0.05% trifluoracetic acid, as reported elsewhere [14]. The peptide concentration in WSE was evaluated by the *o*-phtaldialdehyde (OPA) method, as described by Church et al. [15].

Total free amino acids (TFAAs) were quantified by a Biochrom 30+ series Amino Acid Analyzer (Biochrom Ltd., Cambridge Science Park, Cambridge, UK) with a Li-cation-exchange column (4.6 × 200 mm internal diameter) [14].

### 2.5. Bread Making

Seven experimental breads were manufactured using a type “0” wheat flour (Molino Casillo) with the following components: moisture, 12%; protein, 13.9% of dry matter, d.m; fat, 2.3% of d.m; dietary fiber, 2.2% of d.m; and carbohydrates, 81% of d.m. All breads were obtained from doughs with DY 160, corresponding to a flour/water ratio of 62.5/37.5% (*w*/*w*), and we added 2% wt/wt commercial baker’s yeast (AB Mauri Italy S.p.a., Casteggio, Italia) to all of them. The experimental breads were as follows: W-B, control bread made with wheat flour; HF-B, a bread containing 6% wt/wt of HF DWG in place of wheat flour; f-HF-B, a bread containing 6% wt/wt of f-HF DWG; xf-HF-B, a bread containing 6% wt/wt of xf-HF DWG; LF-B, a bread containing 6% wt/wt of LF DWG; f-LF-B, a bread containing 6% wt/wt of f-LF DWG; xf-LF-B, a bread containing 6% wt/wt of xf-LF DWG.

Doughs were manually mixed for 5 min, divided into 150 g pieces, shaped, and rested in pans for 10 min at 25 °C. Proofing was performed at 28 °C for 1.5 h and a relative humidity of 75%. The leavening performance levels of the doughs were determined and expressed as the volume increase (ΔV, mL) [16]. The breads were baked at 220 °C for 20 min in a professional oven (VEVOR EB-4D, VEVOR, Taicang, China), and then cooled at room temperature for 2 h before analysis.

### 2.6. Bread Characterization

#### 2.6.1. Biochemical and Nutritional Properties

The analysis of the pH, TTA, organic acids, and TFAA of the dough after proofing was carried out as reported above. The proximate composition of the experimental breads was calculated as reported above [11]. For an assessment of the in vitro protein digestibility (IVPD), breads underwent a sequential enzymatic treatment carried out according to the consensus method developed within a large European framework (COST Action InfoGest). The digestion procedure consisted of three consecutive phases, and all the gastro-intestinal fluids were prepared according to the INFOGEST protocol [17]. Briefly, 2 g of homogenized bread was added to 2 mL of simulated salivary fluid (containing 150 U/mL of porcine α-amylase) to create the oral step of the digestion. The bolus was incubated for 5 min at 37 °C and then we added 4 mL of simulated gastric fluid (containing 2000 U/mL of pepsin), reproducing the gastric phase. After 120 min at 37 °C, 8 mL of pancreatic (containing pancreatin, 200 U/mL based on trypsin activity) and bile fluids were added. The chyme was incubated for a further 120 min at 37 °C and, at the end of the digestion, centrifuged for 15 min at 10,000× *g*. The Bradford method [18] was used to determine the concentrations of the protein in the digested and non-digested fractions. IVPD was expressed as the percentage of the total protein solubilized after enzyme hydrolysis.

The starch hydrolysis index of bread (HI) was determined by mimicking the in vivo digestion of starch [19]. Bread portions, containing 1 g of starch, were treated with amyloglucosidase, and the released glucose was quantified with a glucose assay kit (GOPOD-format, Megazyme) following the manufacturer’s instructions. The degree of starch digestion was expressed as the percentage of potentially available starch hydrolyzed after 180 min. Control wheat bread (WB) was used as a reference to estimate the hydrolysis index (HI = 100). The equation pGI = 0.549 × HI + 39.71, proposed by Capriles and Arêas [20], was used to calculate the predicted glycemic index (pGI).

#### 2.6.2. Technological Properties

Bread volume, calculated as the loaf volume/weight ratio, was determined by the rapeseed displacement method 10–05.01 [11]. The crumb grain of breads was evaluated using image analysis technology. Two-dimensional images of 1.5 cm thick bread slices were acquired by flatbed scanning using an image scanner Epson XP-255 (Electronics Ltd., Suwa, Nagano, Japan) at 300 dpi and analyzed in gray scale (0–255). Segmentation was carried out with ImageJ software (National Institutes Health, Bethesda, MD, USA).

The chromaticity coordinates of the crust and crumb of the bread were obtained by a CS-10 colorimeter (CHN Spec Technology, Hangzhou, China) and reported as the color difference, Δ*E*
∗ *ab*, calculated by the following equation:$$\Delta E\;*\;ab=\sqrt{{\left(\Delta L\right)}^{2}+{\left(\Delta a\right)}^{2}+{\left(\Delta b\right)}^{2}}$$
where Δ*L*, Δ*a*, and Δ*b* are the differences for *L*, *a** and *b** values between the sample and a reference (a white ceramic plate with *L* = 92.2, *a** = 0.15, and *b** = 0.85).

#### 2.6.3. Sensory Analysis

Ten trained panelists (five males and five females, mean age: 34 years, range: 25–50 years) were recruited for the sensory analysis. The analyzed attributes included visual and tactile perceptions (color of crust and crumb, elasticity); taste (acidic taste, sweetness, salty, herbaceous taste, bitter flavor); odor perception (acidic, toasted); and chewing (chewiness), which were each rated using a scale from 0 to 10, with 10 being the highest score. Samples were served in random order and evaluated by all panelists in two replicates. Before the sensory evaluation, loaves were cooled at room temperature for 4 h after baking, then cut into slices 1.5 cm thick. Final scores were calculated as the means of each attribute. According to the ethical guidelines of the sensory laboratory, the enrolled panelists, who did not suffer from any food intolerances or allergies, received information on the objectives of this study and provided written informed consent.

### 2.7. Statistical Analysis

DWG sourdoughs and breads were produced in duplicate, and all the microbiological, biochemical, nutritional, technological, and sensory analyses were carried out in triplicate for each batch of samples. Data underwent one-way ANOVA, and pair comparison of treatment means was performed by applying Tukey’s procedure at *p* < 0.05, using the statistical software Statistica 12.5 (StatSoft Inc., Tulsa, OK, USA). The kinetics of acidification were modeled according to the Gompertz equation using the statistical software Statistica 12.5.

The data obtained from the nutritional and technological characterization of fortified breads were also analyzed through partial least-squares discriminant analysis (PLS-DA), using MetaboAnalyst version 5.0 software (metaboanalyst.ca/; accessed online 14 April 2024).

## 3. Results

### 3.1. DWG Fractions

The coarse and fine DWG fractions identified by air classification had significantly different particle size distributions (*p* < 0.05), with D90 results of 492 and 68 µm for LF and HF, respectively (Appendix A). Namely, 62% of particles in HF were between 100 and 1000 µm and 37% of particles were between 10 and 100 µm. On the contrary, LF was characterized by a higher percentage of particles in the ranges of 1–10 and 10–100 µm (48 and 38%, respectively; Appendix A).

HF and LF also had different proximal compositions (Appendix A). Whilst the protein content in LF was 63% higher than that in HF, the latter had a higher fiber content (up to 50 g/100 g d.m.) compared to LF, which resulted in a different soluble/insoluble fiber ratio (Appendix A).

The DWG fractions were also analyzed for their mycotoxin contents. Aflatoxins B1, B2, G1, and G2 were not detected, whereas deoxynivalenol and ochratoxin A were found at around 370 and 1.2 µg/kg, respectively, without any significant differences (*p* > 0.05) between HF and LF. Nonetheless, the values were under the thresholds defined by the European regulations 1881/2006 and 165/2010.

### 3.2. Bioprocessed DWG Characterization

HF and LF were submitted to a biotechnological process, carried out by *L. plantarum* T6B10 and *F. sanfranciscensis* A2S5, alone or in combination with commercial food-grade xylanase. The DWG fractions before treatment were characterized by the presence of low cell densities of yeasts and *Enterobacteriaceae* (on average, 1.48 and 2.73 log cfu/g, respectively), which were not found after treatment. During fermentation, the LAB cell density increased by ca. 2 log cycles, ranging from 9.11 to 9.50 log cfu/g, with no significant differences (*p* > 0.05) among bioprocessed samples (f-HF, xf-HF, f-LF, xf-LF). Molds were not found in any of the samples, whereas yeasts were detected only in HF and LF (2.72 ± 0.13 and 2.75 ± 0.26 log cfu/g, respectively).

Doughs from both DWG fractions had an initial pH of 6.26, which decreased by ca. 2.5 units during the 24 h fermentation. Overall, the fastest acidification was observed in f-LF and xf-LF, in which an average Vmax and λ of 0.37 ΔpH/h and 1.15 h were, respectively, found, as opposed to 0.26 ΔpH/h and 1.63 h in bioprocessed HF (Figure 1). Although a slightly higher rate of acidification and lower latency phase were observed in samples containing xylanase (xf-HF and xf-LF) compared to those fermented without the enzyme, no significant differences (*p* > 0.05) were found.

The TTA values were ca. 26 and 40 mL in bioprocessed HF and LF, respectively. Accordingly, differences in the organic acid content, mostly lactic acid, were observed between the two fractions. Indeed, while f-LF and xf-LF contained up to 84 mmol/kg of lactic acid, f-HF and xf-HF contained ca. 50 mmol/kg. Acetic acid, instead, ranged between 7.78 and 9.49 mmol/kg, with xf-HF and xf-LF showing the highest values.

HF and LF contained different amounts of phytic acid, which was roughly 50% higher in LF compared to HF (Table 1), but regardless of whether there was bioprocessing, the phytic acid decreased by ca. 35% after incubation. Compared to HF and LF, bioprocessing also led to an increase in soluble fibers, by up to 34 and 60% in f-HF and f-LF, respectively. Slightly lower increments were observed when xylanase was used during fermentation; moreover, the total dietary fiber and arabinoxylan contents decreased in all bioprocessed samples, especially those treated with xylanase (Table 1).

### 3.3. Effect of Bioprocessing on DWG Protein

The extent of proteolysis in bioprocessed samples was preliminarily evaluated by referring to SDS-PAGE protein patterns. Before treatment, DWG fractions were characterized by ca. 13 bands evenly distributed within the gel. In HF, those at around 150 and below 10 kDa had the strongest intensity, whereas in LF, the bands between 150 and 250 kDa, as well as those between 30 and 40 kDa, covered over 60% of the total intensity. As shown in Figure 2, bioprocessing strongly modified the protein profile, reducing the intensity of the bands over 150 kDa and increasing that of the bands below 20 kDa, in bioprocessed HF and LF, respectively. Proteolysis in LF also led to the complete disappearance of the bands between 100 and 25 kDa.

A complementary RP-FPLC analysis of the WSE was also performed to understand the modification occurring at MW below 10 kDa, which included peptides poorly separated by SDS-PAGE. Among the DWG fractions, LF had the highest total peak area, almost double that of HF, and they had different distributions of peaks in the chromatogram (Table 2). Bioprocessing strongly decreased the peptide content; indeed, compared to LF, the total area was up to 2-fold lower in f-LF and xf-LF. These data were also confirmed by the quantification of peptides with the OPA method (Table 1). While in HF, the distribution of the peaks barely changed during bioprocessing, in f-LF and xf-LF, an increase in highly hydrophilic peptides (eluting at the lowest concentration of acetonitrile) was observed compared to LF.

As a consequence of proteolysis, a modification of the free amino acid profile was also observed (Figure 3). Before fermentation, LF had a higher amino acid content than HF, but their distributions were similar, with Asn, Glu, Trp, Asp, Ala, Arg, and Pro being the most abundant (ranging from 106 to 548 mg/kg), whereas Cys and Met were the least represented, with concentrations below 25 mg/kg. Fermentation significantly increased the TFAA content, which was more than doubled in bioprocessed samples compared to raw DWG fractions (Table 1). The overall amino acid profile changed after bioprocessing. Indeed, while Asn, Glu, and Arg were still the most abundant amino acids, with concentrations ranging from 472 to 768 mg/kg, significant increases (*p* < 0.05) were observed in others, especially for Leu, Phe, Lys, and Met, which were up to 13-, 10-, 7-, and 11-fold higher compared to the raw DWG fractions. The highest increments were observed in xf-HF and xf-LF (Figure 3). Compared to HF and LF, only Ser decreased (*p* < 0.05) after bioprocessing, while the Cys content increased only in f-HF and xf-HF.

### 3.4. Biochemical and Nutritional Characterization of Breads

With the aim to produce fortified breads, raw and bioprocessed DWG fractions were used as ingredients in bread making. Three control breads were prepared: one containing only wheat flour (WB), and two containing raw DWG fractions, HF and LF (HF-B and LF-B, respectively). Four more experimental breads containing the same amount of bioprocessed DWG were also produced, obtaining f-HF-B, xf-HF-B, f-LF-B, and xf-LF-B. The supplementation significantly affected breads’ proximal composition, increasing the fiber and protein contents (Appendix A). Breads were characterized by protein contents from 7.56 to 8.68 g/100 g, with highest values recorded for breads containing raw or bioprocessed LF. Meanwhile, HF-B, f-HF-B, and xf-HF-B showed the highest fiber contents, reaching 3.71 g/100 g.

The fortification-modified organic acid and TFAA contents, according to the DWG fractions added, are shown in Table 3. Indeed, breads baked with f-LF and xf-LF contained the highest amounts of lactic acid, at almost 19 mmol/kg. Meanwhile, DWG fractions treated with xylanase provided the highest amounts of free amino acids (up to 750 mg/kg).

The nutritional indexes notably varied among the breads (Table 3). WB was characterized by the highest glycemic index, and yet the sole addition of HF or LF decreased the pGI by 5 or 4%, respectively. Further decreases (*p* < 0.05) were observed when breads were fortified with bioprocessed DWG fractions (up to 11%). WB also had the lowest IVPD, which significantly (*p* < 0.05) increased in all experimental breads, reaching 80% in xf-LF-B (Table 3).

### 3.5. Technological and Sensory Properties of Fortified Bread

The presence of raw DWG fractions negatively affected the volume increase after proofing, which equated to 13.73 ± 0.58 and 12.02 ± 0.57 mL in HF-B and LF-B, respectively, as compared to WB (16.60 ± 0.78 mL). Nevertheless, when bioprocessed DWG fractions were used, the volume increase in the doughs before baking was significantly higher compared to WB, reaching 21.75 mL in xf-LF-B. The same trend was observed after baking, when measuring the specific volume of baked loaves (Table 3), as well as from the image analysis of bread slices (Appendix A). Indeed, the percentage of black pixels, indicating crumb porosity, was lower in HF-B and LF-B (16.9 and 14.1%, respectively) and higher in f-HF-B (25.5%), xf-HF-B (26.1%), f-LF-B (18.5%), and xf-LF-B (22.1%), compared to WB (18.2%).

Crust and crumb colorimetric coordinates of the fortified breads, expressed through the color difference index (ΔE), differed according to the bioprocessing employed (Table 3). Compared to WB, significantly (*p* < 0.05) lower values of *L* (lightness) and higher values of *a* (green/red index) characterized xf-HF-B and xf-LF-B crusts. Crumb lightness, on the other hand, did not differ among samples; in this case, *a* increased with the addition of DWG fractions, either raw or bioprocessed.

Fortified breads also underwent a sensory analysis carried out by trained panelists (Figure 4). Breads containing bioprocessed DWG fractions were characterized by higher scores for acidic odor and taste compared to WB, HF-B, and LF-B. xf-HF-B and xf-LF-B, as confirmed by the color analysis, were found to have a more intense crust color, greater toasted notes, and better elasticity.

### 3.6. Multivariate Statistical Analysis: PLS-DA

The overall data collected from the nutritional and technological characterization of the breads were applied to a multivariate statistical analysis aiming at verifying the contribution of air classification and enzymatic and microbial bioprocessing to the quality of breads and highlighting any differences. Partial least-square discriminant analysis (PLS-DA) was carried out on normalized data of breads categorized into three distinct groups (Figure 5A). WB was clearly separated from the fortified breads, while the fortified breads were divided in two almost distinct clouds. The top twenty Variable Importance in Projection (VIP) scores were in the range of 1–1.5, and their impact on sample stratification was supported by cross-validating coefficients. Indeed, the Q2 coefficient (Figure 5B), which tended to R2, indicated the goodness of PLS-DA. The VIP scores (Figure 5C) demonstrated that the variables that mainly contributed to the stratification were the specific volume, crumb and crust color coordinates, predicted glycemic index, IVPD, and the contents of some amino acids, thus highlighting the importance of fractionation and bioprocessing in modeling the quality features of breads.

## 4. Discussion

The food system is currently facing a multitude of challenges, the majority of which are a direct consequence of climate change, land erosion, and the ever-increasing global population. These factors are compelling the food industry to adopt new, sustainable approaches that can effectively meet the rapidly evolving and intensifying demand for food. To ensure the resilient sustainability of the food system, (i) enhancing the agrifood productivity, (ii) promoting healthy and sustainable diets, and (iii) improving food waste and by-product management are among the critical action points necessary, which require us to accelerate technological innovation [21]. Under that framework, this study aimed at developing a bioprocessing protocol for the valorization of DWG, one of the major side streams of the cereal industry, by using two fractions obtained by air classification, and thus fostering its utilization in healthy and sustainable food products.

DWG was micronized and underwent air classification, during which particles are separated according to their aerodynamic properties, resulting in a coarse fraction, often rich in starch and fiber, and a fine fraction, rich in protein [22]. Apart from differences in their particle size distributions, the two fractions obtained from DWG flour differed mainly in their fiber (mainly present in the coarse fraction, HF) and protein (main constituent of the fine fraction, LF) contents. Since it is more energy efficient than other fractionation techniques, dry fractionation has been proposed as a sustainable option to obtain protein-rich flours from cereals and legumes, as well as to modulate their levels of anti-nutritional compounds (e.g., phytic acid, condensed tannins, trypsin inhibitors, etc.) [22]. However, in this study, the protein-rich fraction contained a higher phytic acid content than HF. Phytic acid can bind to proteins, starch, and minerals, changing their solubility, functionality, digestion, and absorption [23]. Since fermentation with selected LAB has been demonstrated to be an effective method for reducing anti-nutritional factors and enhancing the value of cereal industry by-products, *L. plantarum* T6B10 and *F. sanfranciscensis* A2S5 were used to ferment HF and LF. The strains, previously selected for their acidification and proteolytic potential during DWG fermentation [9], were used in this study alone or in combination with food-grade xylanase.

Both fractions proved to be suitable substrates for the growth of LAB, which increased their cell density by ca. 2 log cycles. Microbial contamination from *Enterobacteriaceae* and yeasts in HF and LF was extremely low, as a result of the oil extraction process during which hexane temperatures reached 68 °C, and completely disappeared after fermentation. The strains showed good pro-technological performances and, as a consequence of the abundant production of lactic and acetic acids, acidified the doughs more efficiently than when they were used to ferment unfractionated DWG. Indeed, the lag phase recorded during the acidification kinetics was roughly three times smaller than that observed by Perri et al. [9]. This might have been because of the higher water content used for the doughs in this study. When DWG was micronized prior to fractionation, the surface area per weight and the water absorption rate increased, and DY 200 was not sufficient to hydrate the doughs as it was in the study of Perri et al. [9]; hence, the doughs were prepared with a DY of 350. Moreover, the faster acidification observed in f-LF and xf-LF compared to HF bioprocessed samples could have been related to the presence of more damaged starch, a trait often observed in fractions with smaller granule sizes [24], such as LF. The acidification, as well as the activity of microbial phytases, were also responsible for the decrease in phytic acid content (up to 36% in f-HF and xf-HF), confirming the results obtained when the same starters were used to ferment unfractionated DWG [9]. Phytases of *L. plantarum*, and to a lesser extent *F. sanfranciscensis*, were previously studied and characterized and were found to have a different optimum pH, preferably around 4.0 [25]. In this study, the contribution of endogenous phytases was likely minimal since their optimal temperature is around 50 °C [25,26], and the oil extraction process was conducted at a higher temperature, which might have denaturized them or at least slowed their activity.

One of the main components of DWG fractions is dietary fibers, which are carbohydrates that cannot be digested or absorbed in the small intestine. While soluble dietary fibers are appealing for their ability to lower the blood glucose response and plasma cholesterol levels, impacting the microbial production of short-chain fatty acids at the gut level, insoluble dietary fibers can increase the stool volume and mitigate constipation symptoms [27]. During fermentation, an increase in soluble fibers, accompanied by a decrease in total fibers, was observed (Table 1). Although it is commonly accepted that fermentation affects the fiber content by increasing the amount of soluble fibers at the expense of insoluble ones [27], very little information can be found on the overall balance that this modification affects. This trend was slightly amplified in xf-HF and xf-LF. It is thus hypothesized that the solubilization of the insoluble fraction and the further solubilization of arabinoxylans in DWG fractions treated with xylanase provides more ready-to-use carbohydrates, which are then used for fermentation and not detected as either soluble or insoluble fibers.

The efficient proteolytic system is another typical trait of lactic acid bacteria, which is why during bioprocessing of DWG fractions, proteolysis was deeply evaluated through a combination of electrophoretic and chromatographic techniques. The relatively high amount of proteins, which include albumin, globulin, prolamin, and glutelin, and the well-balanced amino acid profile make DWG a particularly interesting plant-protein source [28]. From the SDS-PAGE emerged the finding that except for three bands between 70 and 120 kDa, which were present in LF and absent in HF, the fractions mainly differed in the intensity of the bands rather than the number. Indeed, high-MW (250 kDa) proteins were extremely pronounced in LF. The polypeptides at 30 and 40 kDa in LF (Appendix A), which might have been components of wheat germ 8S globulin [29], also had a higher intensity compared to HF and, together with the high-MW proteins, were most predominant in the fine fraction. The electrophoretic profile of bioprocessed DWG fractions highlighted extensive hydrolysis (Figure 2). Differences between the two bioprocessing protocols within the same fraction were not observed, implying that the strains were the sole factor responsible for proteolysis and that xylanase did not contribute to an evident protein modification. However, the extent of proteolysis was amplified in f-LF and xf-LF compared to f-HF and xf-HF. Indeed, all the bands between 250 and 10 kDa faded considerably after bioprocessing of LF DWG.

Since SDS-PAGE only retains proteins within the range of 10–250 kDa, the effect of fermentation and enzymatic treatment on peptides below 10 kDa was evaluated through RP-FPLC. Overall, significant differences (*p* < 0.05) in terms of total area and peptide distribution were highlighted by the chromatographic analysis based on hydrophobic interactions between HF and LF. In all bioprocessed samples, the total area was subject to significant decreases compared to untreated samples. In accordance with the electrophoretic results, proteolysis was markedly higher in LF bioprocessed samples. Although the decrease in peptides is not in line with that commonly found during fermentation with lactic acid bacteria, and previous studies on DWG or wheat germ did not focus on the protein profile [9,29], it is easily explained. The LAB proteolytic system includes several peptidases with high specificity for wheat polypeptides [30], which result in a significant proportion of the protein and peptides being converted into amino acids, reducing the peptide yield. To confirm this assumption, the free amino acid profiles of raw and bioprocessed DWG fractions were also analyzed through liquid chromatography. Indeed, the TFAA content, which was almost double compared to that obtained during DWG fermentation with the same strains [9], exceeded 7 g/kg in xf-LF. The TFAA content was also significantly higher (*p* < 0.05) in xylanase-treated samples (xf-HF and xf-LF) compared to those fermented without the enzyme (f-HF and f-LF). It is possible that the arabinoxylan solubilization when the enzyme was used accelerated LAB metabolism, resulting in a faster acidification rate (Figure 1) and more intense proteolysis. A similar trend was observed when wheat bran was submitted to a bioprocessing protocol involving fermentation with selected LAB strains and xylanase treatment [31]. Overall, the amino acid profile of the fractions, which included all essential amino acids, corresponded to that already reported [9,28], and Lys, the major limiting amino acid in wheat flour, significantly (*p* < 0.05) increased after bioprocessing, reaching 500 mg/kg in xf-LF. Amino acids play an important role in several physiological functions including antioxidative responses, neurological development, gene expression, growth, and metabolism of the gut microbiota and many others [32]; hence, their improved bio-accessibility in the bioprocessed DWG fraction is highly advantageous. This is the case for GABA (γ-aminobutyric acid), a non-protein amino acid that acts as a neurotransmitter in the mammalian central nervous system and has different physiological effects, including anti-depressive, antioxidant, and hypotensive effects [33]. The GABA concentration significantly increased in bioprocessed DWG fractions, exceeding 600 mg/kg in f-LF and xf-LF.

To investigate the ability of air classification, as well as enzymatic and microbial bioprocessing, to modify DWG’s potential as a food ingredient, raw and bioprocessed DWG fractions were dried at a low temperature and used to produce fortified breads. Bread is a major staple food widely consumed worldwide, and over the past decades, researchers have focused on improving its quality, studying its nutritional and functional aspects, texture, shelf-life, and so on, and simultaneously responding to the trending demand for clean-label products [34]. The major biochemical characteristics (proximate composition, organic acids, TFAA) of the experimental breads produced in this study changed according to the fraction and the bioprocessing protocol used. When HF was added, the fiber content reached 3.6 g/100 g of bread (Appendix A), thus reaching the threshold required to make the “source of fiber” claim, according to Regulation EC No. 1924/2006. Meanwhile, all breads, regardless of the fraction used, could be labeled a “source of protein” since up to 13% of the energy was provided by proteins. The organic acid and TFAA contents, instead, strictly depended on the bioprocessing used, with these found to be lower in WB, HF-B, and LF-B compared to breads fortified with bioprocessed DWG fractions, and similar to those commonly reported for wheat bread [9]. Furthermore, although the sole addition of HF or LF increased the TFAA concentration compared to WB, further increases were observed in f-HF-B, xf-HF-B, f-LF-B, and xf-LF-B (Table 3).

The supplementation with bioprocessed DWG highly impacted breads’ nutritional properties, increasing the protein digestibility and decreasing the predicted glycemic index. The predicted glycemic index, quantified with a multi-step enzymatic treatment mimicking in vivo digestion [19], estimates the amount of carbohydrates that will raise the glucose levels in the blood once the breads are ingested. pGI was lower in HF-B and LF-B compared to WB, most likely due to the higher contents of fibers and proteins, which are known to influence macronutrient digestion and absorption during gastrointestinal transit, and thus the bread glycemic index [35]. The higher organic acid and soluble fiber contents provided by bioprocessed DWG fractions (Table 1) further decreased pGI in f-HF-B, xf-HF-B, f-LF-B, and xf-LF-B compared to WB and the respective controls. Lactic acid is known to induce interactions between starch and gluten, reducing starch availability, whereas acetic acid is associated with a delay in the gastric emptying rate [36]. Soluble fibers, which were significantly higher in f-HF-B, xf-HF-B, f-LF-B, and xf-LF-B, instead, have viscous properties capable of slowing the glucose absorption rate at the intestinal level [36]. The improved protein digestibility, correlated with the intense proteolysis, is in line with the results obtained by Perri et al. [9] on unfractionated DWG. Nonetheless, when we look beyond the nutritional and functional potential, the amino acids and low-molecular-weight peptides generated during bioprocessing also strictly influenced the aroma profile of the breads. Above all, xf-HF-B and xf-LF-B were characterized by more peculiar features generated during baking (Figure 4), from the acidic odor and taste typical of sourdough breads to the more intense flavor intensity determined by amino acids, like aspartic acid and glutamate, and volatiles.

Although it was suggested that fractionation by air classification could reduce the unpleasant impact on bread’s structure [10], supplementation of HF and LF in this study, due to the lower content of gluten compared to the endosperm and the higher amount of fiber, which interfered with the development of an optimal gluten network [6,7], negatively affected crumb porosity and bread elasticity, thus confirming DWG’s limitations. Nevertheless, these effects were mitigated in f-HF-B and f-LF-B, most likely due to the increase in soluble fibers provided with the supplementation compared to HF and LF, and completely overcome in xf-HF-B and xf-LF-B. Indeed, the degradation of water-insoluble arabinoxylans by xylanase lowers viscosity, making water available for gluten or starch. Moreover, although not visible with the SDS-PAGE performed in this study, xylanases were found to affect protein secondary structures and increase the proportion of β-turns, leading to a better-hydrated gluten network in bread dough [37], which is why the bread structure (volume before and after cooking, crumb cell size and percentage) in xf-HF-B and xf-LF-B visibly improved, not only compared to HF-B and LF-B but also WB (Appendix A). It could also be argued that the xylanase in DWG sourdoughs provided more available sugars, which, combined with the higher amino acid content, enhanced the yeast metabolic activity during fermentation, providing a better development of the doughs (xf-HF-B and xf-LF-B). Yet, reducing sugars and free amino acids in xf-HF-B and xf-LF-B also participated in a more intense Maillard reaction during baking, as observed from the colorimetric coordinates (Table 3). It should be noted that although xylanases’ positive impact on doughs’ extensibility has been reported in several studies, one of the drawbacks frequently observed with their use is an increase in stickiness or bread firming rates if the proper doses are not used [37]. Since xylanases from different manufacturers’ might contain different combinations of enzymes with different enzymatic activities, it is imperative that, for each of those preparations, the dosage and processing parameters are tailored to bakers’ needs.

## 5. Conclusions

To fully exploit cereal industry side streams like DWG, a comprehensive approach must be implemented, including novel and sustainable techniques like air classification, as well as enzymatic and fermentation processes, specifically targeting the structure. DWG has an unparalleled nutritional potential, with exceptional fiber and protein contents as well as phenolic compounds. However, its use in the food industry is still limited.

The approach employed in this study had not previously been explored in the context of DWG fractions. The results obtained indicate that the combination of all the techniques is the optimal method for balancing the nutritional and technological properties of breads. Each approach contributes specific characteristics to the fortified breads. Xylanase is indispensable for achieving excellent textural properties, such as volume and elasticity. Fermentation is crucial for enhancing the nutritional properties, including IVPD and pGI. Air classification can produce fractions with a specific proximal composition, to meet the needs of markets other than for bread, such as protein-rich foods. The incorporation of bioprocessed DWG fractions could be extended to other bakery products, potentially enhancing the nutritional quality of a wide range of food products. Moreover, this process can be easily scaled up, and enzymes commonly used in the baking industry are typically considered processing aids, meaning they do not need to be listed among the ingredients. This bread would, therefore, align with the clean-label concept while, at the same time, contributing to healthy and sustainable diets.

## Figures and Tables

**Figure 1 foods-13-01953-f001:**
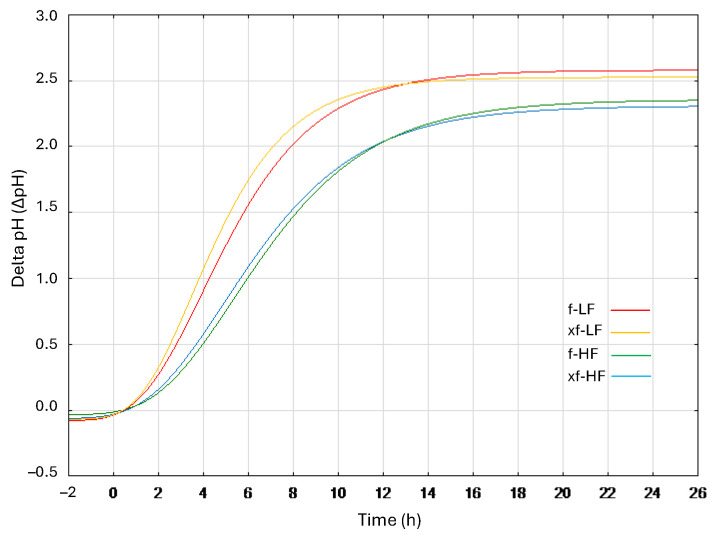
Acidification kinetics of coarse (HF) and fine (LF) defatted wheat germ fractions, during fermentation with *L. plantarum* T6B10 and *F. sanfranciscensis* A2S5 alone (f) or in combination with xylanase (xf).

**Figure 2 foods-13-01953-f002:**
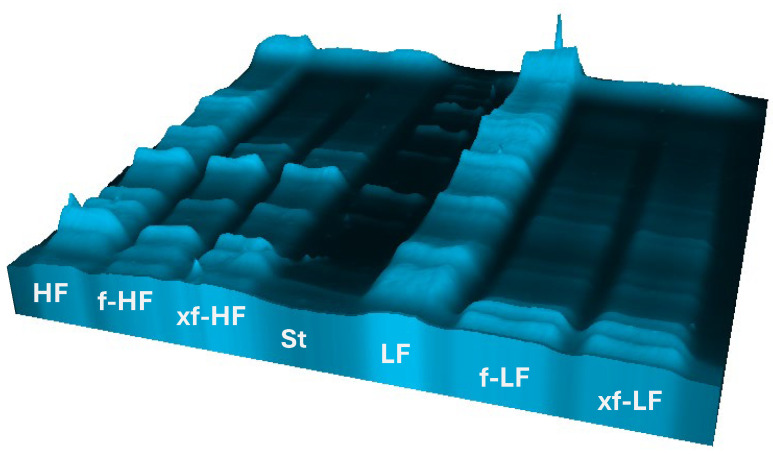
Three-dimensional patterns of total proteins obtained by sodium dodecyl sulfate–polyacrylamide gel electrophoresis (SDS-PAGE) of coarse (HF) and fine (LF) defatted wheat germ fractions, before and after fermentation with *L. plantarum* T6B10 and *F. sanfranciscensis* A2S5 alone (f) or in combination with xylanase (xf). St, protein standard (Bio-Rad, Hercules, CA, USA).

**Figure 3 foods-13-01953-f003:**
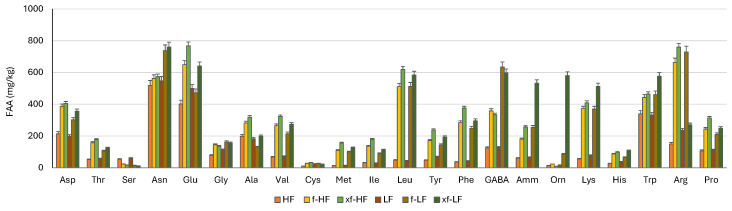
Concentrations of free amino acids (mg/kg d.m.) of coarse (HF) and fine (LF) defatted wheat germ fractions, before and after fermentation with *L. plantarum* T6B10 and *F. sanfranciscensis* A2S5 alone (f) or in combination with xylanase (xf).

**Figure 4 foods-13-01953-f004:**
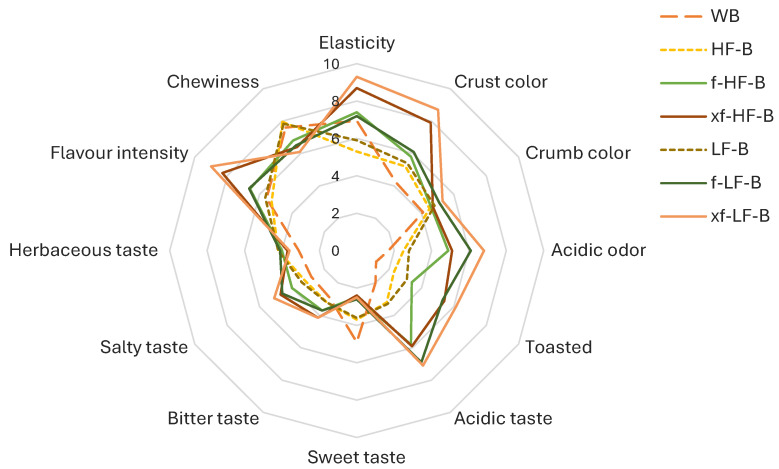
Sensory analysis of breads. WB, control wheat bread. HF-B and LF-B, control breads containing coarse (HF) and fine (LF) defatted wheat germ fractions before and after fermentation with *L. plantarum* T6B10 and *F. sanfranciscensis* A2S5 alone (f-HF-B and f-LF-B) or in combination with xylanase (xf-HF-B and xf-LF-B).

**Figure 5 foods-13-01953-f005:**
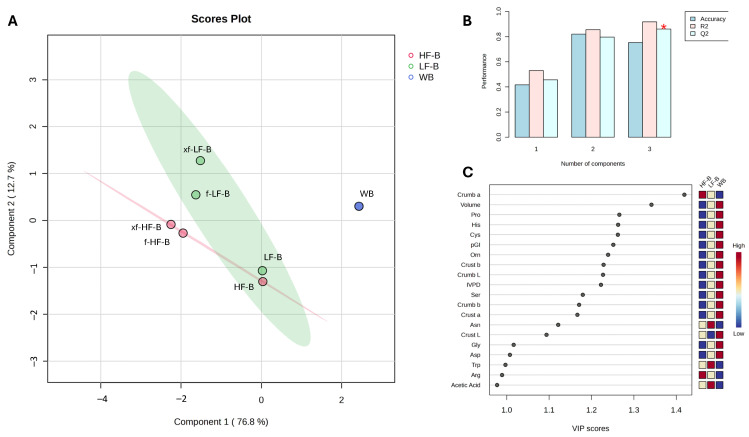
Partial least-squares discriminant analysis (PLS-DA) of breads’ nutritional and technological features. Score plot (**A**), cross-validated Q2/R2 coefficients (**B**), and VIP scores (**C**) resulting from the PLS-DA of control (WB, blue), high-fiber defatted wheat germ (HF, pink), and low-fiber defatted wheat germ (LF, green) breads. WB, control wheat bread. HF-B and LF-B, control breads containing coarse (HF) and fine (LF) defatted wheat germ fractions before and after fermentation with *L. plantarum* T6B10 and *F. sanfranciscensis* A2S5 alone (f-HF-B and f-LF-B) or in combination with xylanase (xf-HF-B and xf-LF-B). Statistical significance (*p* < 0.05) is identified by the presence of an asterisk.

**Table 1 foods-13-01953-t001:** Biochemical characterization of coarse (HF) and fine (LF) defatted wheat germ fractions, before and after fermentation with *L. plantarum* T6B10 and *F. sanfranciscensis* A2S5 alone (f) or in combination with xylanase (xf).

	HF	f-HF	xf-HF	LF	f-LF	xf-LF
Lactic acid (mmol/kg)	0.34 ± 0.01 ^c^	49.38 ± 1.35 ^b^	50.21 ± 1.12 ^b^	0.22 ± 0.06 ^c^	83.54 ± 5.08 ^a^	74.9 ± 4.03 ^a^
Acetic acid (mmol/kg)	0.19 ± 0.04 ^c^	7.78 ± 0.35 ^b^	9.49 ± 0.43 ^a^	0.26 ± 0.01 ^c^	8.6 ± 0.37 ^a^	8.94 ± 0.44 ^a^
FQ	1.77 ± 0.03 ^e^	6.35 ± 0.14 ^c^	5.29 ± 0.09 ^d^	0.86 ± 0.01 ^e^	9.71 ± 0.49 ^a^	8.38 ± 0.33 ^b^
Peptides (mg/g)	55 ± 2.16 ^cd^	50 ± 1.87 ^d^	58 ± 2.10 ^c^	75 ± 4.23 ^a^	57 ± 2.00 ^c^	61 ± 2.33 ^b^
TFAA (g/kg)	2.64 ± 0.04 ^d^	6.08 ± 0.35 ^b^	6.96 ± 0.44 ^a^	2.96 ± 0.19 ^c^	6.05 ± 0.35 ^b^	7.28 ± 0.48 ^a^
Phytic acid (g/100 g)	0.45 ± 0.02 ^c^	0.33 ± 0.01 ^d^	0.34 ± 0.01 ^d^	0.69 ± 0.02 ^a^	0.51 ± 0.02 ^b^	0.52 ± 0.01 ^b^
Total dietary fiber (g/100 g d.m.)	50 ± 0.49 ^a^	38 ± 0.67 ^b^	34 ± 0.61 ^c^	29 ± 1.48 ^d^	24 ± 0.59 ^e^	21 ± 0.47 ^f^
of which soluble fiber (g/100 g d.m.)	5.32 ± 0.42 ^e^	7.14 ± 0.29 ^c^	6.33 ± 0.17 ^d^	7.82 ± 1.03 ^c^	12.51 ± 0.54 ^a^	10.02 ± 0.32 ^b^

The data are the means of three independent experiments ± standard deviations (*n* = 3). ^a–f^ Values in the same row with different superscript letters differ significantly (*p* < 0.05).

**Table 2 foods-13-01953-t002:** Area, number of peaks, and peak area of peptides as determined by reversed-phase fast-performance liquid chromatography (RP-FPLC) of coarse (HF) and fine (LF) defatted wheat germ fractions, before and after fermentation with *L. plantarum* T6B10 and *F. sanfranciscensis* A2S5 alone (f) or in combination with xylanase (xf). Peak area of detected peptides, expressed as a percentage of the total area, is separated based on hydrophobicity (elution at Eluent B with different percentages).

			Peaks Area (%)			
	Area (mAU × mL)	N. Peaks	0% B	0–46% B	46–100% B	100% B
HF	2150 ± 108 ^b^	56 ± 2 ^a^	39 ± 1.67 ^b^	50 ± 2.44 ^b^	2.27 ± 0.23 ^b^	8.79 ± 0.29 ^b^
f-HF	1866 ± 74 ^c^	49 ± 2 ^b^	35 ± 1.79 ^b^	56 ± 2.36 ^b^	0.00 ± 0.00 ^d^	9.23 ± 0.24 ^a^
xf-HF	2093 ± 99 ^b^	57 ± 2 ^a^	37 ± 1.60 ^b^	54 ± 2.50 ^b^	0.00 ± 0.00 ^d^	8.42 ± 0.22 ^b^
LF	4220 ± 142 ^a^	48 ± 1 ^b^	26 ± 1.75 ^c^	64 ± 1.63 ^a^	4.73 ± 0.22 ^a^	5.58 ± 0.27 ^d^
f-LF	1990 ± 126 ^b^	55 ± 2 ^a^	49 ± 1.68 ^a^	39 ± 2.17 ^d^	2.36 ± 0.16 ^b^	9.40 ± 0.31 ^a^
xf-LF	2319 ± 121 ^b^	51 ± 2 ^ab^	48 ± 1.90 ^a^	44 ± 1.16 ^c^	1.17 ± 0.06 ^c^	7.27 ± 0.25 ^c^

The data are the means of three independent experiments ± standard deviations (n = 3). ^a–d^ Values in the same row with different superscript letters differ significantly (*p* < 0.05).

**Table 3 foods-13-01953-t003:** Biochemical, nutritional, and technological characterization of breads. WB, control wheat bread. HF-B and LF-B, control breads containing coarse (HF) and fine (LF) defatted wheat germ fractions before and after fermentation with *L. plantarum* T6B10 and *F. sanfranciscensis* A2S5 alone (f-HF-B and f-LF-B) or in combination with xylanase (xf-HF-B and xf-LF-B).

	WB	HF-B	f-HF-B	xf-HF-B	LF-B	f-LF-B	xf-LF-B
Lactic acid (mmol/kg)	0.27 ± 0.00 ^f^	0.41 ± 0.05 ^e^	13 ± 0.37 ^c^	15 ± 0.02 ^b^	0.63 ± 0.07 ^d^	17 ± 1.03 ^a^	19 ± 0.25 ^a^
Acetic acid (mmol/kg)	0.01 ± 0.00 ^d^	0.13 ± 0.04 ^c^	1.21 ± 0.48 ^a^	1.53 ± 0.31 ^a^	0.45 ± 0.14 ^b^	1.29 ± 0.65 ^a^	1.29 ± 0.52 ^a^
FQ	19 ± 0.02 ^a^	3.09 ± 0.24 ^f^	11 ± 0.26 ^d^	9.87 ± 0.01 ^e^	1.39 ± 0.43 ^g^	13 ± 0.38 ^c^	15 ± 0.14 ^b^
TFAA (mg/kg)	299 ± 11 ^d^	451 ± 19 ^c^	673 ± 26 ^b^	729 ± 24 ^a^	471 ± 19 ^c^	671 ± 26 ^b^	749 ± 22 ^a^
HI (%)	100 ± 1.97 ^a^	90 ± 2.19 ^b^	82 ± 1.96 ^c^	80 ± 2.61 ^c^	94 ± 3.33 ^b^	86 ± 1.08 ^c^	83 ± 2.98 ^c^
pGI	94 ± 1.08 ^a^	89 ± 1.20 ^b^	85 ± 1.07 ^c^	84 ± 1.43 ^c^	91 ± 1.83 ^b^	87 ± 0.69 ^c^	85 ± 1.64 ^c^
IVPD (%)	62 ± 1.55 ^c^	67 ± 2.15 ^b^	73 ± 2.34 ^a^	74 ± 1.91 ^a^	68 ± 2.19 ^b^	76 ± 3.16 ^a^	81 ± 2.32 ^a^
Specific volume (cm^3^/g)	1.62 ± 0.04 ^c^	0.85 ± 0.02 ^f^	1.31 ± 0.04 ^d^	1.84 ± 0.05 ^b^	0.74 ± 0.04 ^f^	1.16 ± 0.04 ^e^	2.41 ± 0.06 ^a^
Image analysis							
Black pixel area (%)	18 ± 0.29 ^c^	17 ± 0.14 ^d^	25 ± 1.01 ^a^	26 ± 0.99 ^a^	14 ± 0.27 ^e^	18 ± 0.31 ^c^	22 ± 0.85 ^b^
Average cell size (pixel)	101 ± 3.82 ^c^	82 ± 0.98 ^d^	102 ± 1.37 ^c^	120 ± 3.43 ^b^	85 ± 1.08 ^d^	126 ± 6.04 ^b^	185 ± 4.21 ^a^
Crust color							
*L*	61 ± 1.98 ^a^	64 ± 6.96 ^a^	64 ± 4.58 ^a^	45 ± 5.61 ^c^	53 ± 1.61 ^b^	62 ± 7.14 ^a^	36 ± 2.8 ^d^
*a*	9.91 ± 0.77 ^c^	8.30 ± 0.61 ^d^	6.51 ± 0.62 ^e^	13 ± 0.45 ^a^	12 ± 0.64 ^b^	10 ± 0.84 ^c^	13 ± 0.24 ^a^
*b*	30 ± 3.78 ^a^	28 ± 1.81 ^a^	29 ± 3.86 ^a^	27 ± 3.6 ^a^	30 ± 3.01 ^a^	31 ± 0.34 ^a^	21 ± 2.62 ^b^
ΔE	44 ± 1.54 ^c^	40 ± 1.60 ^c^	40 ± 1.59 ^c^	55 ± 1.66 ^a^	50 ± 2.17 ^b^	43 ± 2.54 ^c^	61 ± 4.18 ^a^
Crumb color							
*L*	65 ± 1.70 ^a^	60 ± 1.34 ^a^	60 ± 1.36 ^a^	54 ± 1.39 ^b^	61 ± 2.44 ^a^	59 ± 2.56 ^a^	59 ± 4.65 ^a^
*a*	−1.56 ± 0.02 ^f^	0.38 ± 0.02 ^b^	0.47 ± 0.02 ^a^	−0.03 ± 0.01 ^d^	−0.31 ± 0.01 ^e^	0.23 ± 0.02 ^c^	−0.37 ± 0.02 ^e^
*b*	18 ± 0.14 ^b^	20 ± 0.58 ^a^	20 ± 0.54 ^a^	20 ± 0.6 ^a^	19 ± 0.41 ^a^	20 ± 0.48 ^a^	20 ± 0.37 ^a^
ΔE	32 ± 1.57 ^c^	38 ± 4.31 ^ab^	37 ± 2.95 ^ab^	43 ± 3.45 ^ab^	36 ± 1.28 ^b^	38 ± 4.55 ^ab^	38 ± 1.73 ^b^

The data are the means of three independent experiments ± standard deviations (n = 3). ^a–g^ Values in the same row with different superscript letters differ significantly (*p* < 0.05).

## Data Availability

The data presented in this study are available upon request from the corresponding author.

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
