# Peer review of "Effect of Air Classification and Enzymatic and Microbial Bioprocessing on Defatted Durum Wheat Germ: Characterization and Use as Bread Ingredient"

_foods, 2024, doi:10.3390/foods13121953_

Round 1

Reviewer 1 Report

Comments and Suggestions for Authors

The paper describes an interesting set of experiments on potential use of defatted durum wheat germ as an ingredient for nutritional improvement of bread. The obtained data are correctly presented. The results are compared are related to previous similar work. Also, the results are interesting for their potential for practical application. Overall, I consider this to be an interesting paper that should be published. Nevertheless, I have few concerns that should be addressed before the paper can be published.

Specific points that the authors need to address:

Title:

In the title of the manuscript, it should be specified which by-product of durum wheat milling the research deals with. The research did not deal with different by-products but with one by-product that was treated in different ways. Moreover, defatted germ is not a direct by-product from the mill, it is a raw germ, while defatted germ is obtained after removing the oil from it. Even more, the raw germ is not a by-product that is obtained in the largest amount in the mill, it is the bran. Because of all the above mentioned, I am of the opinion that in the title the term „durum wheat milling by-products“ must be replaced with „defatted durum wheat germ“.

 Abstract:

Line 12:  In what context is the term protein quality used? The very next sentence says "negative impact on food structure", which contradicts the previous statement. In the context of bread production, the quality of protein refers primarily to the technological quality that comes from gluten and not the proteins present in the germ (albumins and globulins). Germ proteins have nutritional value and in that context have quality, but this must be clearly stated in the text.

Line 19: Include more specific quantitative results: "free amino acids increased by 2-fold" should be stated as "free amino acids increased from - to  in mg/g".

 Introduction:

Lines 34-36: In what context the term greatest by-product of the milling industry is used? Wheat germ is certainly not that in quantity. The by-product that is obtained in the largest amount is bran, and if the mill is not set to separate the germ as a separate by-product, it also ends up in the bran. Germ is in the share of wheat grain with about 2.5-3%, In addition, it cannot be separated during milling to that percentage and usually the yield of the germ as a separate by-product is only about 0.2-0.3% compared with the yield of bran of around 25%. If "greatest" refers to nutritional quality and the possibility of using it as a nutritionally valuable food supplement, then it should be emphasized as such. Correct the text in accordance with previous comments.

 Lines 58-61: I do not understand why the authors believe that the use of gluten and ascorbic acid is not in line with clean-label products. They cannot be considered undesirable because gluten is already part of wheat flour which is the main raw material from which bread is made, and ascorbic acid (vitamin C) is certainly not undesirable or harmful. If the definition of "clean label products" is viewed as using as few ingredients as possible and ingredients that consumers can use at home, then the use of DWG itself does not fall into that category, and especially not DWG treated with lactic acid bacteria. Please elaborate these issues!

Lines: 61-65: Include some more details about the methods and outcomes of previous studies on how do the different stabilization methods (fermetation, enzyme denaturation, acidification, oil removal) compare in terms of effectiveness and impact on DWG properties?

 Materials and methods:

Line 175: delete brackets at the end of the sentence “81% of d.m.)”.

Control bread: It is a bit unusual to have 3 control samples - in this case 3 control breads. What is compared to what? What is the control's control? Please explain the reasoning behind that.

Lines 183-185: How was variability in the manual mixing process controlled? Were there specific guidelines according to some standardized method or training provided to ensure uniformity across batches?

 Results:

Lines 251-254: This is not supported by the data given in Table S1. From the data on D10, D50 and D90, it is not possible to extract the specified data on the share of fractions of a certain particle size interval. The authors could support the stated data by providing a graph of particle size distribution from Mastersizer 3000.

 Table S1: It would be useful if the authors provide data on the composition of the initial DWG. In this way, it would be shown how the air classification affects the change in the chemical composition of the fractions, that is, what is the enrichment of these fractions in terms of protein or fiber content.

 Discussion:

Lines 443-445: It is not clear to me on what basis the authors conclude that coarse DWG fractions is rich in starch? You can’t refer carbohydrates as starch. Germ is rich in sugars such are sucrose or raffinose even more than starch which is mostly present as a contamination from residual endosperm. Please elaborate or change the text accordingly.

 The discussion mentions potential issues with xylanase dosage leading to increased stickiness or bread firming rates. Can you provide guidance on how to optimize xylanase use to avoid these issues?

 General comment: How do you envision these findings being applied on a larger scale, are there any foreseeable challenges in scaling up these processes?

 Conclusions:

Lines 610-612: It is not usual to comment on something in the conclusions that was not previously mentioned in the text (sustainable milling technologies). Also, it is not clear to me what the authors mean by sustainable milling technologies. Either explain this or leave it out of the text.

Author Response

Revisore 1

The paper describes an interesting set of experiments on potential use of defatted durum wheat germ as an ingredient for nutritional improvement of bread. The obtained data are correctly presented. The results are compared are related to previous similar work. Also, the results are interesting for their potential for practical application. Overall, I consider this to be an interesting paper that should be published. Nevertheless, I have few concerns that should be addressed before the paper can be published.

The authors thank the reviewer for the comments. Each comment has been addressed and a point by-point revision provided.

Specific points that the authors need to address:

Title:

In the title of the manuscript, it should be specified which by-product of durum wheat milling the research deals with. The research did not deal with different by-products but with one by-product that was treated in different ways. Moreover, defatted germ is not a direct by-product from the mill, it is a raw germ, while defatted germ is obtained after removing the oil from it. Even more, the raw germ is not a by-product that is obtained in the largest amount in the mill, it is the bran. Because of all the above mentioned, I am of the opinion that in the title the term „durum wheat milling by-products“ must be replaced with „defatted durum wheat germ“.

The suggestion is highly appreciated and the title has been modified accordingly.

Abstract:

Line 12:  In what context is the term protein quality used? The very next sentence says "negative impact on food structure", which contradicts the previous statement. In the context of bread production, the quality of protein refers primarily to the technological quality that comes from gluten and not the proteins present in the germ (albumins and globulins). Germ proteins have nutritional value and in that context have quality, but this must be clearly stated in the text.

The negative impact on food structure is a combination of both the lower content of gluten compared to the endosperm and the higher amount of fiber (aspect discussed in Lines 50-59; 632-635). The reviewer is correct, we should have been more precise. The term “protein quality” was intended as protein nutritional quality which was previously found comparable to the reference protein defined by FAO/WHO (doi.org/10.1016/j.foodchem.2006.04.040; https://doi.org/10.1016/j.tifs.2018.06.001). The statement, already reported in the introduction (Lines 48-50) was clarified in the abstract (Line 12).

Line 19: Include more specific quantitative results: "free amino acids increased by 2-fold" should be stated as "free amino acids increased from - to  in mg/g".

 Ok. The sentence was revised.

Introduction:

Lines 34-36: In what context the term greatest by-product of the milling industry is used? Wheat germ is certainly not that in quantity. The by-product that is obtained in the largest amount is bran, and if the mill is not set to separate the germ as a separate by-product, it also ends up in the bran. Germ is in the share of wheat grain with about 2.5-3%, In addition, it cannot be separated during milling to that percentage and usually the yield of the germ as a separate by-product is only about 0.2-0.3% compared with the yield of bran of around 25%. If "greatest" refers to nutritional quality and the possibility of using it as a nutritionally valuable food supplement, then it should be emphasized as such. Correct the text in accordance with previous comments.

The reviewer is correct and since it is not easy to retrieve the exact amount of durum wheat germ, let alone defatted durum wheat germ generated, we used wheat germ percentage of all the kernel, to give an idea of the volume of by-product produced. The sentence has been clarified.

Lines 58-61: I do not understand why the authors believe that the use of gluten and ascorbic acid is not in line with clean-label products. They cannot be considered undesirable because gluten is already part of wheat flour which is the main raw material from which bread is made, and ascorbic acid (vitamin C) is certainly not undesirable or harmful. If the definition of "clean label products" is viewed as using as few ingredients as possible and ingredients that consumers can use at home, then the use of DWG itself does not fall into that category, and especially not DWG treated with lactic acid bacteria. Please elaborate these issues!

Although there is not a legal definition for clean label foods, and they are open to different interpretations, it is generally acknowledged that a “clean label” product contains natural ingredients and does not include over-processed compounds. The concept of “clean” label foods includes products that are free of ingredients that consumers will find unacceptable like synthetic additives, preservatives, artificial colors, saturated fatty acids and hydrogenated lipids (https://doi.org/10.3390/foods11182796 ). For instance, ascorbic acid is an additive used in foods as ascorbate salts, which depending on the type are classified as E300-E301-E302-E304 by the EU Reg. 1333/2008 and labeled as such, thus it falls in the category of ingredients that consumers might find unacceptable. Gluten is not considered a food additive according to the same regulation, but it would be the umpteenth over-processed ingredient in a long list of ones. Indeed, although it seems unlikely that a food product, that could ideally be prepared with just three ingredients (flour, water, leavening agent), could contain that many ingredients, this is not the case. Store-bought breads contain dozens of ingredients which include several types of flour, water, leavening agents, sweeteners, shorteners, improvers, preservatives, vitamins and so on (Smith, E., Benbrook, C., & Davis, D. R. (2012). A closer look at what’s in our daily bread. The Organic Center). Hence, since the clean label concept also applies to short ingredients lists, not only easy to read and minimally processed ingredients, any addition which does not constitute a core ingredient could jeopardize the clean label status of that product. This aspect was implemented in the text (Lines 68-70).

Lines: 61-65: Include some more details about the methods and outcomes of previous studies on how do the different stabilization methods (fermetation, enzyme denaturation, acidification, oil removal) compare in terms of effectiveness and impact on DWG properties?

 Ok. A brief explanation has been added (Lines 39-44).

Materials and methods:

Line 175: delete brackets at the end of the sentence “81% of d.m.)”.

Done.

Control bread: It is a bit unusual to have 3 control samples - in this case 3 control breads. What is compared to what? What is the control's control? Please explain the reasoning behind that.

The control bread composed of wheat flour was necessary to evaluate the effects of DWG fractions on bread and to demonstrate that their addition worsens bread technological properties, which we could only hypothesize when the experimental plan was designed. Similarly, HF-B and LF-B were used as controls for the breads containing bioprocessed fractions because it was necessary to discriminate the effect of the treatment over that of the ingredient.

Lines 183-185: How was variability in the manual mixing process controlled? Were there specific guidelines according to some standardized method or training provided to ensure uniformity across batches?

The manual mixing was performed by the same operator according to an internal straight-dough procedure that ensured uniformity across batches. Each type of bread was mixed for 5 minutes to reach homogeneous consistency and smooth surface, divided into pieces, shaped, and rested in pans for 10 min before proofing.

Results:

Lines 251-254: This is not supported by the data given in Table S1. From the data on D10, D50 and D90, it is not possible to extract the specified data on the share of fractions of a certain particle size interval. The authors could support the stated data by providing a graph of particle size distribution from Mastersizer 3000.

The information provided in the text was calculated based on the particle size distribution from Mastersizer 3000. The suggestion was highly appreciated. Graphs of particle size distribution have been added to the supplementary material and the statement in the text clarified (Lines 270-271)

Table S1: It would be useful if the authors provide data on the composition of the initial DWG. In this way, it would be shown how the air classification affects the change in the chemical composition of the fractions, that is, what is the enrichment of these fractions in terms of protein or fiber content.

Ok. The initial DWG composition was included in table S2

Discussion:

Lines 443-445: It is not clear to me on what basis the authors conclude that coarse DWG fractions is rich in starch? You can’t refer carbohydrates as starch. Germ is rich in sugars such are sucrose or raffinose even more than starch which is mostly present as a contamination from residual endosperm. Please elaborate or change the text accordingly.

There must have been a misunderstanding. We did not mean to say that DWG coarse fraction is rich in starch. A brief explanation, which included its own reference, had been used to specify to the reader how air classification works and what its products usually are (Lines 487-489). Indeed, in the next sentence (Lines 489-492), we defined the differences among DWG fractions: fiber and protein content. Starch was never mentioned with regard to DWG, because as the reviewer pointed out, it is not the only carbohydrate in DWG and we did not study the carbohydrate profile of our fractions.   

The discussion mentions potential issues with xylanase dosage leading to increased stickiness or bread firming rates. Can you provide guidance on how to optimize xylanase use to avoid these issues?

The optimization of the xylanase dosage and processing parameters is essential to avoid these issues, unfortunately, it is strictly related to the type of enzyme used since different manufacturers offer commercial preparations that contain a combination of enzymes with different enzymatic activities. This aspect was clarified in the text (Lines 654-657)

General comment: How do you envision these findings being applied on a larger scale, are there any foreseeable challenges in scaling up these processes?

The scale-up of this process is rather simple and closely related to that of sourdough, which is increasingly used at industrial level. Hence the challenges related to it large-scale use are anything but insurmountable. Having industrial fermenters automatic stirrers in the vessel, which provide more oxygenation as well as well as a more surface of contact among microorganisms, flours and water, the fermentation process can be subjected to differences in acidification time. Nevertheless, this is easily fixable by including in the process a pH sensor and a cooling system that automatically refrigerates the dough when the desired pH is reached. The only difference, between a common sourdough and the DWG fermented dough, is the presence of the xylanase, which could continue to work if the dough is left on stall between productions. This issue can be fixed by dehydrating the dough and turning it into a Type III sourdough, which can be stored for long periods of time in an “inactivated” form, thus facilitating bread production processes as well as allowing small bakeries to use bioprocessed DWG as ingredient.

Conclusions:

Lines 610-612: It is not usual to comment on something in the conclusions that was not previously mentioned in the text (sustainable milling technologies). Also, it is not clear to me what the authors mean by sustainable milling technologies. Either explain this or leave it out of the text.

As already introduced in lines 492-494, air classification was proposed a sustainable technique to obtain protein-rich flours from cereals and legumes and it could also be considered a novel technique since, although it has been studies for years from the academic community, applications at industrial level are not that common, especially for DWG. For this reason, in the conclusion this concept was redrawn, by referring to air classification as a “novel and sustainable milling techniques”. Nevertheless, the sentence has been clarified for better interpretation.   

Reviewer 2 Report

Comments and Suggestions for Authors

Introduction

Line 33: I understand this is relevant on a Italian context, Please provide a global context regarding durum wheat production by product.

Line 56-65: Please make the research gap more clear

Materials and methods

Please organize section 2.1 into different subsections:

2.1 Materials, content on Lines 81:84

2.2 Milling and air classification, content of lines 84:92

2.3 DWG characterization, content of lines 93: 100

Line 178: why 6%?

Results

Figure 1 – please correct the x and y legend, what is T ? is it temperature or time? If its time please include the units, the same for DpH – I believe there is room to write the whole word;

Table 1, 2 and 3 – please correct the number of significant figures, for example

Peptides (mg/g)

55.28 ± 2.16cd

49.64 ± 1.87d

58 ± 2.10c

75.17 ± 4.23a

57.18 ± 2.00c

61.42 ± 2.33b

Should be:

Peptides (mg/g)

55 ± 2.16cd

49 ± 1.87d

58 ± 2.10c

75 ± 4.23a

57 ± 2.00c

61 ± 2.33b

Discussion

It would be helpful to use subheadings to organize the discussion into clear sections

Line 462: was the microbial contamination actually tested? Or this is an assumption?

Line 591: why does it affect the gluten network. Any reference to support?

Line 608: Is there any chance of assessing the bread's long-term stability since the LAB could produce certain peptides that could work as preservatives?

Conclusions

So you think this could be translated into other bakery products?  Maybe introducing a comment like this: "The incorporation of bioprocessed DWG fractions into bread formulations can be extended to other bakery products, potentially enhancing the nutritional quality of a wide range of food products."

Author Response

Introduction

Line 33: I understand this is relevant on a Italian context, Please provide a global context regarding durum wheat production by product.

Ok. Data on global durum wheat production have been added.

Line 56-65: Please make the research gap more clear

Ok. Few sentences were added (Lines 60-70).

Materials and methods

Please organize section 2.1 into different subsections:

2.1 Materials, content on Lines 81:84

2.2 Milling and air classification, content of lines 84:92

2.3 DWG characterization, content of lines 93: 100

Ok. Paragraph 2.1 has been reorganized

Line 178: why 6%?

6% is the amount of DWG used in a previous study (Perri et al., 2022), hence in this study the same amount was used to discriminate any potential effect deriving from the air classification process and the biotechnological protocols employed. The amount is enough to fortify a common white bread, providing fibers and proteins.

Results

Figure 1 – please correct the x and y legend, what is T ? is it temperature or time? If its time please include the units, the same for DpH – I believe there is room to write the whole word;

Ok. Figure 1 has been revised.

Table 1, 2 and 3 – please correct the number of significant figures, for example

Peptides (mg/g)

55.28 ± 2.16cd

49.64 ± 1.87d

58 ± 2.10c

75.17 ± 4.23a

57.18 ± 2.00c

61.42 ± 2.33b

Should be:

Peptides (mg/g)

55 ± 2.16cd

49 ± 1.87d

58 ± 2.10c

75 ± 4.23a

57 ± 2.00c

61 ± 2.33b

Ok. Tables have been revised.

Discussion

It would be helpful to use subheadings to organize the discussion into clear sections

Ok. New paragraphs have been added to the discussion.

Line 492: was the microbial contamination actually tested? Or this is an assumption?

As reported in Lines 134-143 (materials and methods) and 287-293 (results), a microbiological characterization of DWG fraction was performed before and after bioprocessing. Hence the statement it is not an assumption but rather a fact supported by results.

Line 591: why does it affect the gluten network. Any reference to support?

The higher content of fibers compared to refined wheat flour and the lower content of proteins responsible for the tenacity, elasticity and extensibility of the gluten network are the cause of the poor technological performances of DWG, relegating it to products like cookies where a scarce development of a gluten network is less detrimental. This aspect, already reported in lines 50-59, was specified in lines 634-636.

Line 608: Is there any chance of assessing the bread's long-term stability since the LAB could produce certain peptides that could work as preservatives?

There is definitely a possibility that the shelf-life of the experimental breads is impacted by LAB fermentation, either through the release of bioactive peptides, or through the production of organic acids with antimicrobial activity. Nevertheless, this would require a whole new study specifically designed to assess potential bioactivities and the identification of the compounds responsible for such activity.

Conclusions

So you think this could be translated into other bakery products?  Maybe introducing a comment like this: "The incorporation of bioprocessed DWG fractions into bread formulations can be extended to other bakery products, potentially enhancing the nutritional quality of a wide range of food products."

Yes, we believe bioprocessed DWG could be used as ingredient in several production processes. We appreciate the suggestion. The sentence was included in the conclusions.

Round 2

Reviewer 1 Report

Comments and Suggestions for Authors

It appears that the authors made an honest attempt to correct the manuscript. My concerns have been satisfied by the authors in their revision. It is now acceptable for publication.